# The Adaptation and Validation of the Trans Attitudes and Beliefs Scale to the Spanish Context

**DOI:** 10.3390/ijerph19074374

**Published:** 2022-04-05

**Authors:** Miguel Ángel López-Sáez, Ariadna Angulo-Brunet, R. Lucas Platero, Oscar Lecuona

**Affiliations:** 1Department of Psychology, Rey Juan Carlos University, Avda/Atenas, s/n, Alcorcón, 28922 Madrid, Spain; lucas.platero@urjc.es (R.L.P.); oscar.lecuona@urjc.es (O.L.); 2Faculty of Psychology and Education Sciences, Universitat Oberta de Catalunya, Rambla del Poblenou, 08018 Barcelona, Spain; aangulob@uoc.edu

**Keywords:** transgender, transphobia, transnegativity, homonegativity, sexism, psychological skills

## Abstract

This article examines the reliability and validity of the adaptation of the Trans Attitudes and Beliefs Scale (TABS), an instrument capable of detecting transphobic positions, to the Spanish context. A total of 829 psychology students participated in the adaptation procedure. A confirmatory factor analysis was performed to study the fit of the new scale to the factor structure of the original scale (interpersonal comfort, gender identity beliefs, and human value). Convergent validity evidence showed significant correlations and predictive levels with different constructs and sociodemographic variables. The internal consistency of the mean scores was adequate at the global level. The study showed that the TABS is a psychometrically sound instrument for the assessment of attitudes toward trans people, particularly in the context of debates over access to rights and the lack of professional training in disciplines such as psychology.

## 1. Introduction

At a time such as the current one, in which Spain is immersed in a social, political and psychological debate on the expansion of the rights of trans people, we propose the adaptation and validation within this context of a scale for the detection of transphobia: the Trans Attitudes and Beliefs Scale (TABS). To that end, the sections that follow briefly contextualize the proposal.

### 1.1. The Social and Legal Context for Spanish Trans People

It has been 30 years since the murder of Sonia Rescalvo in 1991 in Barcelona, Spain. The judicial sentence in the case of this trans woman, who was brutally beaten to death, established the first outlines within the Spanish state of what is today known as transphobia [1]. In the last 15 years, important and positive social and legislative changes have been made in Spain [2,3] to address the discrimination suffered by trans people. In fact, one such change is currently underway in the form of a new proposed law regarding the rights of trans individuals [4]. However, these changes coexist with growing attitudes of intolerance toward this group [5]—especially among trans-exclusive groups [6,7,8], far-right groups, and Catholics [9,10]—that resulted in an almost 10% increase in hate crimes in 2019 [11].

### 1.2. Transphobia in Psychological Sciences

These overt hate incidents are based on beliefs that shape negative attitudes toward trans people. The difficulty in recognizing such attitudes socially and establishing mechanisms for their prevention has historically been reinforced by the narratives created in psychology and psychiatry well into the 20th century, which normalized hatred in order to “correct” those considered inverted, deviant, sick, or deranged [12]. For example, during the 1930s, psychology prescribed the “right” way to be male or female, initiating a new approach to diagnosis and intervention [13]. It was not until the 1960s and 1970s that some voices began to consider the possibility of connections between trans malaise and social imaginaries [12]. However, in the 1980s, the general discomfort around trans people was assigned to trans individuals themselves, and transsexuality was established as a diagnosed disease [12]. Since then, there have been several diagnostic nomenclatures for trans people [14,15], but few focus on the negative attitudes toward trans individuals that are so often the source of their discomfort.

In view of this, social psychology has focused on societal responsibility, proposing to study the beliefs that support an entire regime of abuse and discrimination toward individuals who diverge from cisheteronormativity [16]. The definition of this term has been accompanied by a search for instruments to measure the phenomenon. There are currently as many as 83 general measurement scales [17] and six specific measurement scales to that end [18]. Of these, the most validated to date are the Genderism and Transphobia Scale (GTS), the Transphobia Scale, the Attitudes Toward Transgendered Individuals Scale, and the Transgender Attitudes and Beliefs Scale (TABS) [18]. In fact, the first definition of transphobia comes from the 2002 introduction to the GTS, when Hill described it as a “motivating force for negative reactions to transgendered people that involve fear and disgust in the observer” [19] (p. 199). Later, the anxiety and fear linked to the suffix ‘phobia’ came under debate—the use of the term can limit the assessment of a wide range of cognitions, effects, and inferences—and it began to be replaced by “negativity.” However, most literature on the subject now uses the terms “transphobia” and “transnegativity” interchangeably, incorporating the rejection of the trans universe from a broad perspective.

In 2005, Hill and Willoughby [20] included the repudiation of trans and gender dissident people (masculine women, feminine men, cross-dressers, transsexuals, transgenderists) in the definition of transphobia in their development of one of the few sound psychometric tools available at the time, the GTS. These authors also probed the connections between sexism and anti-transgenderism, concluding that sexist beliefs are a dimension of anti-trans sentiments because they reinforce binary thinking and punish anyone who does not adjust to the sociocultural expectations for their assigned gender identity. The Spanish adaptation of the GTS [21] stressed this idea, in addition to the close relationship between homophobic and transphobic attitudes. Until recently, the GTS was the only instrument adapted for the Spanish population, in a shortened version that reduced the original 32 items to 12, a reduction justified on the basis of crossed loadings in different factors, semantics, ambiguity, weight, and variance interpretation [21]. However, this was a significant reduction in the representation of items related to cognitive and affective components, with six items for each and, therefore, for the corresponding Genderism/Transphobia factor.

In similar reductions, Tebbe et al. [22] eliminated 10 items, but cautioned that the Genderism and Transphobia-Revised-Short Form Scale was less sensitive when measuring more subtle biased attitudes. The tool’s factorial structure was also controversial. Although the Spanish validation indicated the existence of a clear two-correlated factor structure, some authors have produced other more or less reductionist designs, without reaching a factorial solution that explains the differences between the studies [21,22] or the problems with inter-factor discriminant validity [23]. Additionally, some reviews have expressed doubts about the representativity of the contents, since neither cis nor trans experts were consulted about the scale items in order to produce a more detailed final product [17,18]. Another common criticism finds problems in the cultural features, which contain inconsistencies that point to intercultural differences between the samples [22], and the potential influence of factors such as religiousness or advances in civil rights is not taken into account [24]. These shortcomings are especially important in the current Spanish context, where advances in rights and opposition to them are often related to Catholic beliefs and ideals [25,26].

### 1.3. The Trans Attitudes and Beliefs Scale (TABS)

Considering these limitations, the option provided by Kanamori et al. [24] is particularly promising. They proposed a scale, the TABS, that is sensitive to religious (particularly Christian) nuances, going beyond negative attitudes and exploring the variability of attitudes in different dimensions. Additionally, the TABS recognizes civil rights debates and can capture the variability of subtle negative attitudes. Its conceptualization of the attitude construct is multidimensional, better accounting for complexities and taking into account possible cognitive appraisals and affective reactions toward trans people. The dimensions incorporated into the TABS are: (a) interpersonal comfort (TABS-IC), that is, deriving pleasure from interacting and relating to trans persons in different more or less close contexts, whether formal or informal; (b) beliefs beyond the binary around gender and sexual identity (TABS-BG), that is, the recognition of other realities within the range of diversity and the possibility of exercising rights by belonging to them; and (c) the value of recognizing humanity (TABS-HV), that is, viewing trans persons as human beings, and thus considering them intrinsically and idiosyncratically valuable subjects by virtue of the fact that they are people. The TABS is a 29-item scale (TABS-IC with 14 items; TABS-BG with 10 items; and TABS-HV with 5 items), and higher scores reflect more favorable attitudes toward trans persons. Regarding its psychometric robustness, for this study, evidence of internal structure was obtained through an Exploratory Factor Analysis (EFA) and a Confirmatory Factor Analysis (CFA) for a model of three correlated factors, with adequate goodness-of-fit indices for the latter [24]. Regarding reliability, the α estimates ranged from 0.93 to 0.97 and the test-retest for the three subscales ranged from 0.62 to 0.77. Although there is no evidence of unidimensionality, the findings indicate an overall α of 0.98 [24].

Both this study [24] and subsequent studies establishing the relationships between the TABS and other variables [27] qualify some of the factors associated with transphobia: religiousness (especially religious fundamentalism) as an ideology based on the guardianship of binary and monoheterosexual gender-related values, clearly threatened by trans people; contact, particularly the absence of intergroup contact with trans people, which is associated with and predicts lower acceptance and support, as well as greater prejudice and discrimination; and gender identity and sexual orientation, particularly being male and heterosexual. Likewise, these connections (by findings of correlates, predictors, or significant differences) are replicated not only in the general population, but also among healthcare workers. As Kanamori and Cornelius-White have shown, health professionals tend to have more positive attitudes, a pattern replicated by women [28]. Similar results have been found in the few studies that have used the TABS in the Spanish context among psychology students, adding nuances that influence anti-trans attitudes such as the constructs of sexism, homophobia, and biphobia, as well as sociodemographic variables such as a right-wing ideology, being male, heterosexuality, and a lack of heterodissident contacts [16]. These contributions provide a more global view of attitudes toward trans people among health professionals that is shared by studies using other instruments [29].

The specificities most often found when testing the convergent validity of the TABS and other instruments are especially relevant in the Spanish context, where little has been written on trans issues and individuals. Moreover, the existing work raises questions about a lack of professionalism among health care providers who treat trans people [30,31,32], despite their particular needs [33,34,35]. Neither does the Spanish collegiate organization have its own guidelines on accompanying trans persons, while psychology study regulations [36] contain no stipulations in this regard.

### 1.4. Current Study

Given the reality of this situation, characterized on the one hand by an absence of literature and guidelines and, on the other, by a lack of analysis of the attitudes held by professionals, the TABS is clearly the appropriate instrument for the Spanish context. It is particularly suitable in light of the country’s current social and political debate on the expansion of rights, the historical delay in the conceptualization and recognition of the existence of hatred and violence against trans people, and the Catholic religious influence that permeates belief systems in Spain. Against this background, the five aims of this study are to (1) adapt the Trans Attitudes and Beliefs Scale (TABS) and investigate its factor structure; (2) provide evidence of its internal structure; (3) explore its internal consistency; (4) gather evidence of its convergent validity; and (5) assess the attitudes held by psychology students in relation to trans people/issues. Since this is an exploratory study, which seeks to intersectionally elucidate the construct of transphobia among little-studied populations in the Spanish context, no hypotheses were made beforehand.

## 2. Materials and Methods

### 2.1. Participants

A sample of 829 psychology majors from the three on-site public universities in Madrid—Universidad Complutense (*n* = 398), Universidad Autónoma de Madrid (*n* = 328) and Universidad Rey Juan Carlos (*n* = 103)—took part in the study. Of them, 79.0% (*n* = 655) were ciswomen.

### 2.2. Procedure

The participants were selected using a stratified random sampling with a confidence level of 95%, maximum variability, and a maximum error of ±3% out of a total population of 3745 students majoring in psychology. The groups from each level were selected at random, establishing a similar sample for first-, second-, third-, and fourth-year students. The selection of the participants followed proportional criteria according to gender identity. If one person declined to participate, another was randomly selected. The rejection or nonresponse rate of the individuals selected was 30%. All the participants were informed of the voluntary nature, confidentiality, and anonymity of their responses.

To translate the scale, versions of the items adapted by three experts in gender psychology were used, after which a back-translation (a translation from English into Spanish by a professional Spanish translator in this field, followed by a translation back into English by a professional American translator) was done to avoid significant semantic differences between the translation and the original. At this point, and to address some of the criticism levelled at the original tool and its validity [37], the set of items was reviewed by (a) two experts in psychometry; (b) two trans experts in trans and non-binary topics; and (c) a pilot group of eight psychology students. These reviews helped improve the nuances of each item, and their comprehension and clarity. As a result of these contributions, the term “transgender” was replaced by “trans,” a more recent coinage that is more inclusive with regard to the various identities and experiences involved in the act of crossing boundaries.

### 2.3. Measures

All the scales except for the sociodemographic questionnaire and the Social Desirability Scale used a 6-point Likert scale (1 = strongly disagree; 6 = strongly agree) in order to avoid neutral answer trends and to homogenize the survey information.

#### 2.3.1. Questionnaire including Sociodemographic Aspects

Participants reported their gender identity (0 = cisgender man; 1 = cisgender woman; [other options given were not selected]), sexual orientation (1 = heterosexual; 2 = bisexual; 3 = homosexual; 4 = open response option [the open response option was not selected]; recoded as 0 = heterosexual 1 = LGB), age, and academic year (1st, 2nd, 3rd, 4th).

The religious items asked about affiliation (1 = Atheist; 2 = Agnostic; 3 = Christian; 4 = other, such as Jewish, Muslim, Buddhist; 5 = open response option; recoded as 0 = atheist/agnostic; 1 = religious).

The contacts variables asked about the possibility that the participants knew any LGBT individuals (0 = no; 1 = yes).

For political affiliation, a single item was used [38], based on a 4-point Likert scale (1 = left; 2 = center-left; 3 = center-right; 4 = right) and the political affiliation variable appears as “political affiliation.”

#### 2.3.2. Trans Attitude and Belief Scale (TABS)

The psychometric properties of the original version were developed previously, and those of the adapted version are shown in the Section 3. The items are listed in Appendix A in Table A1.

#### 2.3.3. Genderism and Transphobia Subscale-Revised (GTSS-R)

Only 17 items were selected from the Genderism and Transphobia subscale. Higher scores reflect more favorable attitudes toward trans people. The Tebbe et al. [22] GTS model fitted the data acceptably (CFI = 0.92, RMSEA = 0.07 [0.06, 0.08], SRMR = 0.06). The α for the subscale scores of the GTSS-R were 0.94 and 0.86, respectively. In this sample the evidence of internal structure was obtained using a one-factor CFA (CFI = 1.00, TLI = 1.00, RMSEA = 0.03 [0.02, 0.04]) and adequate internal consistency reliability coefficients (ω = 0.90, α = 0.89) were also obtained.

#### 2.3.4. Ambivalent Sexism Inventory-Short Version (ASI)

This version consists of 12 items to assess sexism using two subscales measuring hostile sexism (ASI-HS) and benevolent sexism (ASI-BS). Higher scores reflect more negative attitudes toward women and femininity, as well as stereotypical and binaristic concepts of gender. The original model had an acceptable fit (CFI = 0.91, TLI = 0.90, RMSEA = 0.077 [0.07, 0.09]) and a good α coefficient (ASI-HS, α = 0.85; ASI-BS, α = 0.80). In this study, evidence of internal structure validity was found using a two-factor model (CFI = 0.99, TLI = 0.99, RMSEA = 0.04 [0.04, 0.05]) and its internal consistency reliability is adequate for hostile sexism (ω = 0.84, α = 0.90), with values similar to those of Rollero [39] (ω = 0.66, α = 0.82).

#### 2.3.5. Modern Homonegativity Scale (MHS)

This 24-item scale measures contemporary negative attitudes toward gays and lesbians. It has a unifactorial structure that duplicates its items to handle possible differences in negativity with respect to homosexual men (MHS-G) or women (MHS-L). Higher scores reflect more negative attitudes toward homosexuality. Morrison and Morrison [40] reported a very good overall reliability, with an α of 0.93. The unidimensional scale obtained good goodness-of-fit indices (GOFI) (CFI = 0.957, TLI = 0.953) and excellent internal consistency coefficients (ω = 0.90, α = 0.92).

### 2.4. Data Analyses

The analyses were done using R software. First, the data were described in order to assess the item distribution and choose the best estimator to perform the CFA. Following the recommendations of Viladrich et al. [41] and considering that a 6-point Likert scale with ceiling effects was being used (see the Section 3 for a description of the item distribution), polychoric correlations and the unweighted least squares estimator (ULS) were chosen. To assess the GOFI, a comparative fit index (CFI) and a Tucker-Lewis index (TLI) greater than 0.95, with a root mean error of approximation (RMSEA) less than 0.05, were considered excellent, and a CFI and TLI greater than 0.90 and RMSEA less than 0.08 considered adequate [42,43]. The CFA was performed using the lavaan package [44] and the reliability of the sum scores was assessed with ω-categorical [45], providing α for comparison with previous research. A value of 0.70 was considered adequate reliability. As the factor scores were not affected by the internal consistency reliability, the results were provided with factor scores and with sum scores (again, for comparison with previous research). A value of 0.70 in consistency reliability can be seen as acceptable [46].

Additionally, evidence of convergent validity was provided in bivariate correlations and a multiple linear regression analysis was performed for each dimension of the TABS. Specifically, the correlations for each dimension of the TABS were estimated with each dimension of the GTSS-R, the ASI, the MHS, and the sociodemographic variable questionnaires. After that, the correlations with a potential effect size (>|0.3|, which indicates approximately 10% of the common variance) were then introduced into the regression analysis. The TABS factors were assigned as dependent variables, and the variables with significant correlations above 0.3 were assigned as independent variables.

## 3. Results

### 3.1. Descriptive Statistics

The mean age was 20.8 (SD = 4.02, Mdn = 20, Range = 17 to 60). Of the participants, 71.9% reported being heterosexual, 23.3% bisexual, and 4.8% homosexual; 13.4% reported being religious, with the most common religious affiliation being Catholic (21.5%). Of the Catholics, 75% actively practiced their religion to some degree.

Table 1 shows the means according to sexual orientation (grouping heterosexuals and LGB) and within this, the existing gender identities (male and female).

Figure 1 presents the item distribution in detail. All the items were recoded prior to the analysis, and a high score means low transphobia. As seen in the percentage of “totally agree” responses (number 6), all the items have ceiling effects: a majority of the participants totally agreed with all the statements. This is also seen in the lack of variance (low standard deviation) in all the items. This is particularly severe with some items, where more than 90% of the participants agreed with the statement (see, for example, T22 and T26). This is the case with four of the TABS-HV items.

### 3.2. Evidence of Internal Structure

In order to assess the internal structure of the TABS, three compatible models were assessed according to the underlying theory: a three-factor correlated model, a one-dimensional model, and a bifactor model. Figure 2 shows the standardized factor loadings and correlations between the factors for the three-factor correlated model. The factor loadings were high and similar (homogenous) in the three factors: TABS-IC (0.62 to 0.91), TABS-BG (0.53 to 0.92), and TABS-HV (0.54 to 0.92). The correlations between the three factors were strong (0.65 to 0.75). An adequate GOFI was obtained for the proposed model (CFI = 0.99, TLI = 0.99, RMSEA = 0.06 [0.05, 0.06]).

The strong correlations in the proposed model could indicate a lack of discrimination between the factors. This unidimensionality was also assessed in this sample. In the one-factor model, factor loadings ranged between 0.45 and 0.82. Although the CFI (0.96) and TLI (0.96) were adequate, the RMSEA was unacceptable (0.10 [0.10, 0.10]). With this in mind, a third model (bifactor model) was employed with a general factor and three specific factors. This model obtained excellent GOFI (CFI = 1.00, TLI = 1.00, RMSEA = 0.04 [0.04, 0.04]). A detailed view of this model indicates that the factor loadings for the specific factors are heterogenous between the items, meaning that once the variance has been captured by the general factor, little variance is shared between the items of the same specific factor.

### 3.3. Internal Consistency of the Sum Scores

Table 2 presents descriptive statistics for the sum scores and internal consistency reliability (i.e., ω-categorical [derived from the 3-correlated factor CFA] and α). Regarding the reliability of these scores, considering ω, positive evidence was found for TABS-BG. TABS-IC had a value below 0.70, but close to it. On the other hand, TABS-HV had a totally unacceptable value. Taking α into consideration, the reliability of TABS-IC and TABS-BG can be considered adequate, with TABS-HV more aligned with Kanamori et al. [26]. However, the value of internal consistency for human values did not improve substantially. Considering that α is computed based on Pearson’s correlations, the low values can be considered normal. In fact, if α is computed with the polychoric correlation matrix, the value of internal consistency is 0.84, which is also compatible with Kanamori’s findings.

### 3.4. Evidence of Validity Based on the Relationship with Other Variables

To examine the based evidence of validity related to other variables, the relationship with the total GTS score was first examined. As seen in Table 3, in the case of TABS-IC and TABS-BG, positive evidence of convergent validity was found, obtaining correlation values of 0.72 and 0.74, respectively. On the other hand, TABS-HV has a more moderate correlation.

Secondly, the relation with other nearby variables and sociodemographic variables was examined. For both TABS-IC and TABS-BG, moderate to strong relations were found with the other theoretical scales, with the exception of ASI-BS, which in both cases had values between low and moderate. TABS-HV, in turn, showed low relations with the studied scales. Of the sociodemographic variables, the only notable relationship was that the more leftist political orientation (PA) corresponded with more favorable attitudes on TABS-IC and TABS-BG.

If the regression model (see Table 4) is evaluated for the dependent TABS-IC variable, the adjusted R is 0.29. ASI-BS negatively predicts the TABS-IC (B = −0.10, *p* < 0.001) and the MHS (B = −0.26, *p* < 0.001) scores. On the other hand, in TABS-BG, the MHS negatively predicts BG (B = −0.54, *p* < 0.001), with an adjusted R of 0.34. Finally, ASI-BS (B = −0.4, *p* = 0.03) and the MHS (B = −0.8, *p* > 0.001) negatively predict TABS-HV, with an adjusted R of 0.06.

## 4. Discussion

As noted in the Section 1, this study was structured around five objectives, with the primary goal being to adapt the Trans Attitudes and Beliefs Scale to the Spanish context using psychometrically sound evidence. While future work will track the performance of the TABS with more diverse samples from the Spanish population, the current study nonetheless provides an introduction to the evaluation of attitudes about trans people. With regard to the second goal, concerning internal structure evidence, the confirmatory analysis corroborated the original three-factor correlated model, and the 29 items had adequate factor loadings in the three dimensions. In general, these results are in line with those of Kanamori et al. [24].

Regarding the relation between the dimensions of the TABS, which was the third focus of the study, the correlations were strong. However, a unidimensional model was discarded. The correlation between TABS-IC and TABS-BG suggested that a satisfactory relationship with trans people could indicate a syncretic view of gender identity definitions. Additionally, comfort with contact and proximity seemed to demonstrate some recognition of the value of the people with whom the participants interacted, as indicated by the correlation between TABS-IC and TABS-HV. Finally, and to a lesser extent, the correlation between TABS-HV and TABS-BG indicated that respect for the trans world involved a certain plural perspective regarding identities and their components. However, some studies have indicated that lower correlations in the TABS-HV dimension could be due to dissonance between maintaining binary beliefs and discomfort with interacting with and appreciating trans people [47]. Additionally, the internal consistency coefficients obtained suggested adequate reliability for TABS-IC and TABS-BG. However, both the ω-categorical and the α for TABS-HV indicated some limitations in the reliability of the TABS-HV factor (ω = 0.30). This low reliability could be explained by the low variance of the items in the factor, which in turn produced low variability in the average score. This lack of variability could be due to the fact that the vast majority of the sample is not religious, and these items proposed an assessment according to religious dogma.

Regarding the objective related to obtaining evidence about the other variables, first, favorable convergent validity evidence was found with the other transphobic construct measure (GTSS-R). This indicated that this subscale (GTSS-R) could be used to measure some dimensions of transphobia, but not the extension covered by the TABS. Moreover, the interrelationship with similar constructs showed that sexist (the ASI) and homonegative (the MHS) attitudes negatively correlated with all the dimensions of the TABS, and that some were good predictors of those dimensions; this was coherent with the results found in studies with similar constructs [16,20,21]. In fact, theoretically, all the constructs shared common roots with transphobia [48], by reinforcing an ideology that reproduced a specific status quo and established a single linear correspondence matrix between the biological and the corporal, gender identity, gender expression, and sexual orientation as valid. The fact that trans people find themselves outside the assigned gender identity explained the incongruence or deviation from this matrix. The correlation between the TABS and the sociodemographic variables showed that a right-wing political tendency was the most important sociodemographic variable. This is consistent with earlier findings [16,49,50].

Religiousness is linked to political conservatism, as Kanamori’s detailed and prolific work in this area has shown. The greater the degree of religiousness—chiefly Christian—the more negative the correlation with the TABS dimensions, particularly TABS-IC and TABS-BG. This was consistent with the fact that adherence to Christian religious values is usually related to greater transphobia, since it involves believing in a dualistic world view and holding positions that are resistant to trans rights [27]. The correlations between the TABS dimensions and the variable related to lack of contact showed how this absence negatively influenced attitudes toward trans individuals [51,52], especially with respect to the relational comfort factor (TABS-IC).

The results of this study indicated that having acquaintances was associated with more comfort during contact (TABS-IC) and using fewer stagnant, pro-rights categorizations (TABS-BG). In this respect, some studies emphasize that familiarity with the LGBT world fosters allyship [53]. The low correlational intensity may have been due to the fact that it was the degree of the contact—and not the contact alone—that cultivated the sense of alliance [54].

The participants’ academic year had no significant or very weak correlations, as found in previous studies where clearer connections were only established when the student was receiving specialized training [52]. The results related to the correlations between the variables of gender identity and sexual orientation indicated a trend, in the sense that LGB people and women had more positive attitudes toward trans people, in line with the literature on the subject [27,55]. Although this result must be taken with caution, due to the characteristics of the sample, the result was not surprising; the heterosexual male subject personifies power, and these beliefs uphold the status quo and maintain the matrix of intelligibility, allowing the heterosexual male to hold on to his position at the top of the pyramid.

Finally, with regard to the fifth and final goal, the study found more favorable attitudes toward trans people among people on the left of the political spectrum. Accordingly, certain identities and affiliations may be a risk factor that could endanger proper professional performance, particularly with trans patients. However, as found by similar studies [51,52], the extreme scores (the average scores of the three factors were above the median) and the reliability of the internal consistency (see Table 1 and Table 2) indicated that psychology students generally had favorable attitudes toward trans individuals. Despite the fact that the lack of variance in this tool can be seen as a psychometric weakness, from a psychological point of view, the results are encouraging: psychology students in Madrid have low transphobic attitudes. Attitudes seem to be changing positively compared to the first studies with health providers [56,57].

Thus, the findings concomitant with transphobia are not only the result of statistical analyses but are also coherent with the contributions from the “matrix of intelligibility” from critical theory on sexuality and queer theory. In other words, a person whose sexual character is not the only determinant of their gender identity and sexual orientation risks being seen as unintelligible, unstable, and noncoherent [58,59]. Cis and heterosexual individuals enjoy a normative status within what is defined as valuable, and try to safeguard themselves against any threat that challenges their privileged position [36]. Hence, being a cis, white, religiously and politically conservative male is often underpinned by the rejection and subordination of all things different [60,61], configuring a hegemonic identity where what is correct, natural, and healthy aligns [13].

After this discussion, it is important to qualify the importance of the TABS for its practical implications. Tools such as this one are essential in the field of psychosocial education as well as other disciplines. This is even more so, when recent reports have warned about how trans people perceive discriminatory treatment and poor preparation of professionals regarding the trans universe [31,32,62] and the normalization of microaggressions in therapeutic practice, including disapproval, discomfort, pathologization, and invasiveness [63]. This in turn results in people who need it most—because of their extra exposure to minority stress [33,64,65] and the consequences derived from it [34,35,66]—not going to their health care providers because they do not consider them safe spaces. The consequences of this stigmatization comprise a public health problem. The social support of psychosocial health professionals not only forms part of professional expertise, but may also have a direct impact on the stressors and damage affecting trans individuals [67]. Consequently, health professionals must be familiar with the trans phenomenon and show a level of commitment to the wellbeing of their trans patients if they are to work competently [68]. In this way, the TABS can be used to a diagnostic screening instrument to determine which professional and personal abilities are best suited to healthy and inclusive professional intervention.

## 5. Conclusions

The TABS offers an important multidimensional measurement of attitudes toward trans people in the field. However, additional work is required. For instance, the behavior of the TABS needs to be investigated using more a heterogeneous probability sampling. These findings can be applied to psychology students in the public university system in Madrid, but not to the entire country or the private education system. Expanding the pool could improve the variability of the sample significantly, since the population in this study was clearly feminized, left-leaning, and secularized. In this respect, secularization requires special analysis, since its growth in Spanish society [25] could complicate the use of the TABS in the Spanish cultural context. Similarly, the feminization of the sample may have influenced the differences found with respect to the original scale. Another area for further research concerns the psychometric properties of the TABS in relevant professional populations such as healthcare workers. On the other hand, the response scale was simplified to six items, diverging from the original formulation. In that respect, future studies should reconsider this scale. Given the low variability, it might even be advisable to reduce it to improve the response process. Finally, future studies should not be limited to cross-sectional quantitative data, but must explore the differences between more subtle, implicit attitudes.

## Figures and Tables

**Figure 1 ijerph-19-04374-f001:**
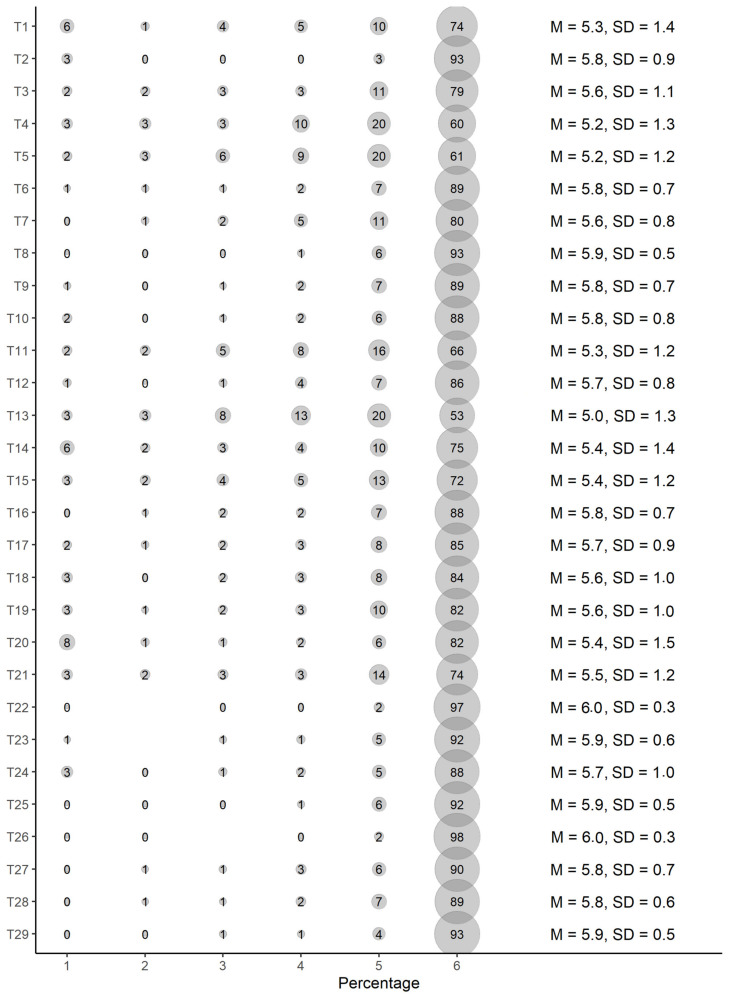
Frequencies and Mean and Standard Deviation for each Variable.

**Figure 2 ijerph-19-04374-f002:**
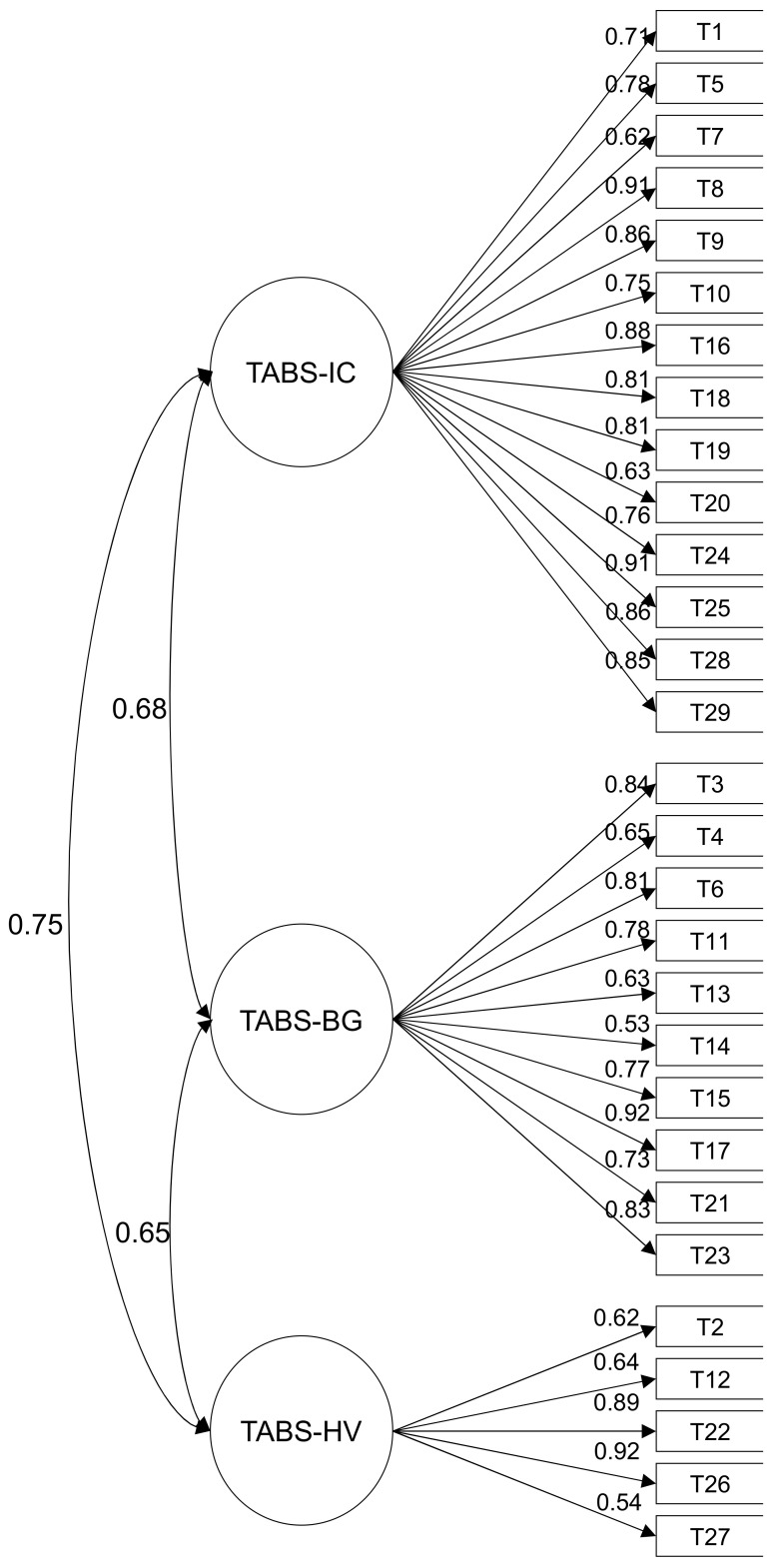
Standardized Factor Loadings for the TABS.

**Table 1 ijerph-19-04374-t001:** Means and Standard Deviations by Gender Identity and Sexual Orientation.

	Heterosexuals				LGB			
	Men		Women		Men		Women	
	M	SD	M	SD	M	SD	M	SD
Trans comfort (TABS-IC)	5.37	0.81	5.68	0.47	5.69	0.50	5.86	0.22
Trans beliefs (TABS-BG)	5.14	0.95	5.48	0.72	5.54	0.65	5.63	0.46
Trans value (TABS-HV)	5.81	0.37	5.86	0.34	5.76	0.54	5.90	0.24
Transphobia (GTSS-R)	5.47	0.75	5.76	0.40	5.74	0.49	5.91	0.16
Hostile sexism (ASI-HS)	2.15	0.99	1.53	0.63	1.39	0.54	1.30	0.50
Benevolent sexism (ASI-BS)	2.26	0.98	1.98	0.70	1.99	0.71	1.76	0.60
Lesbian negativity (MHS-L)	2.35	1.06	1.86	0.77	1.49	0.64	1.33	0.42
Gay negativity (MHS-G)	2.40	1.02	1.95	0.76	1.55	0.58	1.42	0.41
Political affiliation	2.04	0.87	1.96	0.83	1.42	0.63	1.42	0.65
Religiousness	0.23	0.42	0.29	0.46	0.03	0.19	0.13	0.34
No LG contact	0.07	0.25	0.01	0.14	0.01	0.14	0.01	0.10
No B contact	0.12	0.33	0.04	0.21	0.02	014	0.01	0.11
No T contact	0.33	0.47	0.29	0.46	0.15	0.36	0.20	0.40

Note. Political affiliation: 1 = left; 2 = center-left; 3 = center-right; 4 = right. Religiousness: 0 = Not religious; 1 = Religious. No LG contact/No B contact/No T contact: 0 = No contact; 1 = contact.

**Table 2 ijerph-19-04374-t002:** Sum scores and internal consistency reliability.

Variable	M	SD	Mdn	Sk	k	ω	α
TABS-IC	5.67	0.52	83	−2.8	11.4	0.67	0.83
TABS-BG	5.47	0.73	57	−2.5	7.8	0.80	0.86
TABS-HV	5.85	0.35	30	−3.1	11.7	0.30	0.38

**Table 3 ijerph-19-04374-t003:** Bivariate correlations for the study.

Variable	1	2	3	4	5	6	7	8	9	10	11	12	13	14	15	16
TABS dimensions																
1. TABS-IC	1															
2. TABS-BG	0.50 ***	1														
3. TABS-HV	0.38 ***	0.27 ***	1													
Theoretically related scales																
4. GTS-R	0.72 ***	0.74 ***	0.33 ***	1												
5. ASI-HS	−0.40 ***	−0.39 ***	−0.20 ***	−0.49 ***	1											
6. ASI-BS	−0.35 ***	−0.27 ***	−0.18 ***	−0.38 ***	0.46 ***	1										
7. MHS-L	−0.51 ***	−0.58 ***	−0.25 ***	−0.65 ***	0.63 ***	0.42 ***	1									
8. MHS-G	−0.51 ***	−0.58 ***	−0.25 ***	−0.64 ***	0.65 ***	0.4 ***	0.95 ***	1								
Sociodemographic variables																
9. PA	−0.33 ***	−0.30 ***	−0.14 ***	−0.36 ***	0.43 ***	0.32 ***	0.52 ***	0.52 ***	1							
10. REL	−0.20 ***	−0.20 ***	−0.07 *	−0.24 ***	0.19 ***	0.26 ***	0.33 ***	0.33 ***	0.44 ***	1						
11. LLGC	−0.16 ***	−0.11 ***	−0.07 *	−0.16 ***	0.13 ***	0.06	0.16 ***	0.16 ***	0.04	0.04	1					
12. LBC	−0.21 ***	−0.10 ***	−0.06	−0.15 ***	0.12 ***	0.12 ***	0.2 ***	0.2 ***	0.12 ***	0.07	0.25 ***	1				
13. LTC	−0.10 **	−0.06	−0.06	−0.07 *	0.03	0.05	0.17 ***	0.15 ***	0.11 **	0.02	0.22 ***	0.36 ***	1			
14. AY	0.02	0.07 *	0.01	0.02	−0.03	−0.09 *	−0.07	−0.05	−0.05	−0.05	0.00	0.04	0.03	1		
15. GI	0.20 ***	0.14 ***	0.09 *	0.22 ***	−0.25 ***	−0.14 ***	−0.17 ***	−0.18 ***	−0.02	0.07 *	−0.09 **	−0.10 **	−0.01	0.00	1	
16. SO	0.18 ***	0.12 ***	0.02	0.16 ***	−0.21 ***	−0.14 ***	−0.33 ***	−0.33 ***	−0.3 ***	−0.18 ***	−0.05	−0.11 **	−0.11 ***	−0.09 *	−0.03	1

Note. * *p* < 0.05. ** *p* < 0.01. *** *p* < 0.001. 1. TABS-IC = Interpersonal comfort; 2. TABS-BG = Beliefs regarding gender identity; 3. TABS-HV = Human value; 4. GTSS-R = Genderism and Transphobia; 5. ASI-HS = Hostile sexism; 6. ASI-BS = Benevolent sexism; 7. MHS-L = Modern homonegativity-Lesbian; 8. MHS-G = Modern homonegativity-Gay; 9. PA = Political affiliation; 10. R = Religiousness; 11. LLGC = Lack of LG contact; 15. LBC = Lack of bisexual contact; 16. LTC = Lack of trans contact; 17. AY = Academic year (1–4); 18. GI = Gender identity; 19. SO = Sexual orientation.

**Table 4 ijerph-19-04374-t004:** Multiple linear regression models for each dimension of the TABS.

	B	CI95%		*t*	*p*
		Il	Ul		
DV: Trans comfort (TABS-IC)					
Intercept	6.46	6.34	6.56	129.55	<0.001
Hostile sexism (ASI-HS)	−0.04	−0.09	0.02	−1.22	0.23
Benevolent sexism (ASI-BS)	−0.10	−0.15	−0.05	−4.27	<0.001 ***
Modern homonegativity (MHS)	−0.26	−0.31	−0.21	−9.68	<0.001 ***
Political affiliation (PA)	−0.03	−0.07	0.01	−1.43	0.15
DV: Trans beliefs (TABS-BG)					
Intercept	6.45	6.34	6.57	11.27	<0.001
Hostile sexism (ASI-HS)	−0.01	−0.08	0.07	−0.23	0.82
Modern homonegativity (MHS)	−0.54	−0.61	−0.46	−14.90	<0.001 ***
Political affiliation (PA)	0.01	−0.05	0.06	0.25	0.81
DV: Trans human values (TABS-HV)					
Intercept	6.11	6.05	6.19	166.89	>0.001
Hostile sexism (ASI-HS)	−0.02	−0.06	0.03	−0.82	0.41
Benevolent sexism (ASI-BS)	−0.04	−0.07	0.00	−2.15	0.03 *
Modern homonegativity (MHS)	−0.08	−0.12	−0.05	−4.35	>0.001 **

Note. ** p* < 0.05; ** *p* < 0.01; *** *p* < 0.001.

## Data Availability

The raw data supporting the conclusions of this article will be made available by the authors, without undue reservation, to any qualified researcher.

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
