# Peer review of "The Adaptation and Validation of the Trans Attitudes and Beliefs Scale to the Spanish Context"

_ijerph, 2022, doi:10.3390/ijerph19074374_

Round 1

Reviewer 1 Report

Hello, I have attached a document that I hope will be helpful.
Best wishes on your revisions/rewrite.

Author Response

We would like to begin by thanking both the editorial team and the reviewers for taking the time to review our study. We have taken all the comments into account in the course of revising and improving our manuscript. With particular regard to a comment made by Reviewer 1, the original article was reviewed by a native English translator with many years of experience working with academic articles and gender psychology before we submitted it. Nonetheless, in response to the review, the article has been revised once again, paying particular attention to the sections that discuss the quantitative results.

All changes have been marked in yellow within the manuscript.

Introduction:

  1. In addition to incorporating or clarifying some specific points discussed below, we addressed and merged some general suggestions made by Reviewer 1 and Reviewer 2, shortening the introduction (as requested by Reviewer 1), but without losing the social context (as requested by Reviewer 2).

  1. In accordance with Reviewer 2, we have added a short paragraph to highlight the influence of activism in demanding depathologization in medical treatment access: Using the diagnostic category of ‘gender dysphoria’, then, further stigmatizes the trans population and institutionalizes a pathologizing perspective of trans individuals in a way that is currently highly criticized [62], although at the time it was defended by activists as a way to gain access to medical treatment [68].

  1. In response to Reviewer 1, the following clarification regarding the term ‘phobia’ has been included: The earliest studies focused on discovering the attitudes of health personnel towards surgical procedures with trans individuals [38, 31]. It was not until 2002 that the term ‘transphobia’ was defined as a “motivating force for negative reactions to transgendered people that involve fear and disgust in the observer” [42] (p. 199). Later, the anxiety and fear linked to the suffix ‘phobia’ came under debate – the use of the term can limit the assessment of a wide range of cognitions, affects, and inferences – and it began to be replaced by ‘negativity’. However, at present most of the literature on the subject uses the terms ‘transphobia’ and ‘transnegativity’ interchangeably, incorporating the rejection of the trans universe from a broad perspective. In 2005, Daryl Hill and Brian Willoughby [43] included the repudiation of trans and gender dissident people (masculine women, feminine men, cross-dressers, transsexuals and transgenderists) in the definition of transphobia in their development of one of the few solid psychometric tools available at that time.

  1. As requested by Reviewer 2, the following paragraph has been modified to ensure clarity of meaning: Other authors have compiled additional factors associated with more favorable attitudes, such as having progressive political ideologies, belonging to groups that have experience with stigma, or being more knowledgeable because they are young [14].

  1. Similarly, as requested by Reviewer 2, the following paragraph has been clarified as follows: These results are supported by the latest European report, “A long way to go for LGBTI equality” [29], which also highlights how trans people often prefer not to reveal their trans identity to health professionals, even when they feel discouraged and depressed.

  1. Following the recommendations of Reviewer 1, additional information has been provided about the TABS, expanding the conceptual definition of the three scales and explaining the importance of the context. This section now reads as follows: In summary, five key features characterize trans attitudes in the Spanish context. First, Spain is immersed in a social and political debate over the extension of rights to trans people. Second, the country has historically taken a long time to conceptualize and recognize the existence of hatred and violence against trans people, which is still fostered in current circumstances. Third, Spain is a country where the Catholic religious influence has permeated and continues to permeate belief systems. Fourth, there is little literature on the construct of transphobia in Spanish psychology as a field, despite the consequent discomfort for trans persons. Finally, there is only one adaptation of the transphobia measure to Spanish, which is, moreover, deficient in the articulation of the items with respect to the points. Bearing in mind the limitations of other tools, the option provided by Kanamori et al. [52] is particularly valuable and relevant in the Spanish context. They propose a scale that is sensitive to religious (particularly Christian) nuances by going beyond negative attitudes and exploring the variability of attitudes in different dimensions. Additionally, it is receptive to civil rights debates, and is able to capture the variability of subtle negative attitudes. It also conceptualizes the attitude construct in a multidimensional manner, better accounting for the complexities of the construct and incorporating a broad lens that takes into account possible cognitive appraisals and affective reactions toward trans people. The dimensions incorporated by the TABS are: (a) interpersonal comfort, i.e., deriving pleasure from interacting and relating to trans persons in different more or less close contexts, whether formal or informal; (b) beliefs beyond the binary around gender and sexual identity, i.e., the recognition of other realities within the range of diversity and the possibility of exercising rights by belonging to them; and (c) the value of recognizing humanity, i.e., viewing trans persons as human beings, and thus considering them intrinsically and idiosyncratically valuable subjects in themselves by virtue of the fact that they are people. Finally, the psychometric strength was superior to that of other instruments, with an optimal internal consistency with alphas ranging between .96 and .98 in the scale construction studies.

  1. As requested by Reviewer 1, a paragraph has been included to clarify the goal of this study: In light of the above, the aim of this study is (1) to adapt the Transgender Attitudes and Beliefs Scale (TABS) and investigate its factor structure; (2) to provide evidence of its internal structure; (3) to explore its internal consistency; (4) to gather evidence of its convergent validity; and (5) to assess the attitudes held by psychology students in relation to trans people/issues.

  1. In response to Reviewer 1, it is important to emphasize that our aim with this study did not include making a comparison with other instruments; rather we wanted to provide evidence of convergent validity with them. As explained on pp. 16 and 17 of the Standards for Educational and Psychological Testing (AERA, et al. 2014), one of the ways to establish the validity of scores is by evaluating the relationship with similar constructs. In this case, our hypothesis is that the two constructs are related. However, our perspective differs from that of Reviewer 1 regarding the idea that the GTS-R could be used instead of the TABS in our case. Specifically, we argue that while the GTS-R has been established as a short test to evaluate these attitudes, it only has one dimension. The TABS, on the other hand, discriminates between the three dimensions discussed in this work. In this context, it is important to remember that the lack of more specific hypotheses is due to the fact that this is an exploratory study. Moreover, a comparison between the TABS and the GTSS-R is included in the article.

Method:

Participants:

  1. In response to the concerns of Reviewer 1 regarding the representativeness of the sample, we would like to note that our study includes a representative sample of the type of university students who study psychology in Madrid. It is true that this is a sample of young people and that it is feminized, but it reflects the reality of the population of psychology students in Madrid. During the selection of the sample, every effort was made to be scrupulous about the representativeness in percentage terms in accordance with the existing gender identity and course of study. However, if the editorial board believes that it is important to put this in the context of the overall Spanish population, we can change this to make it more in line with the aims of the journal.

Procedure:

  1. In response to Reviewer 1’s comment about the 30% refusal rate, this does not indicate a bias in the sample, since the participants were selected at random. Similarly, the reasons for refusing were quite varied: arriving late or missing the day that the tests were carried out, expressly declining to participate, etc.

  1. In response to Reviewer 1, a definition of ‘double translation’ has been included.

Measures:

  1. In response to Reviewer 1, the response scales were homogenized so as not to encumber the response task while also avoiding central tendency errors. To avoid random responses, control items were included where one of the options could be specifically chosen. In any case, we should expand upon a couple of points. As can be seen in Figure 1, there is a clear ceiling effect in the responses of the participants. Including more points on the scale would merely highlight this lack of variability. Moreover, although changes were made to the format, as noted, a variety of validity and reliability evidence has been provided that attests to its performance. In other words, although the format was changed, the evidence of psychometric quality shows that this did not pose a problem. In fact, the evidence suggests that, for the population of psychology students in Spain, the response scale could be further reduced, since not all six points were used. This may be due to the pronounced asymmetry of the items, with responses outside the low categories appearing infrequently.

  1. In response to Reviewer 2, there were no trans participants, information that we have now included in the section that describes the variables. Moreover, in response to Reviewer 1, a 0 versus 1 coding scheme was chosen instead of 1 versus 2 in order to make a direct interpretation in the correlation table. Similarly, regarding sexual orientation, no participant chose the open response. We have clarified this in the text. Regarding the recoding of the religious variable, we understand that there may be differences between atheists and agnostics, but both groups may be less influenced by the religious social imaginary, either because they do not identify with a religious tradition or because they reject it. Regarding the contact variables, it has been noted that variables are for each particular group. We understand the limitations inherent in the contact variable configured in this way, and now clarify this in the discussion section as follows: The low correlational intensity may have been due to the fact that it was the degree of the contact – and not the contact alone – that cultivated the sense of alliance [58].

  1. In response to Reviewer 1, the precise meaning of the higher scores for each scale have been defined.

  1. In response to Reviewer 1, a note on the existence of an appendix containing the scale items has been included.

Data Analysis

  1. With respect to the comment by Reviewer 1: “As a reader, I want to know more about ‘... polychoric correlations and the unweighted least squares estimator (ULS) were chosen’. I would recommend the authors expand upon this. Yes, they were following Viladrich et al. but what was the reason (or were the reasons), what do these mean - if there were not ceiling effects, what would have been chosen and why. Do polychoric correlations specifically resolve the problem of ceiling effects (e.g., bins within high levels of positivity)? From what I can find, polychoric correlations are based on an assumption that the latent values follow a normal distribution - is this meant as a normal distribution within the very skewed, non-normal manifest items? A bit more explanation would be valuable here”,

we believe that it is not within the purview of this article to provide lessons about the best data matrix to use or which estimator is correct. Interested readers can find more information about these questions and factorial analysis at http://www.psicothema.com/pii?pii=4715

The polychoric correlation matrix is used when the data are ordinal or categorical. In this case, although we have six response categories, there is a marked ceiling effect and the polychoric correlation matrix makes it possible to adjust for this. The fact that it assumes that the latent variables follow a normal distribution does not mean that the variable itself must follow it (of course, it is used for categorical data, with which it is, in point of fact, impossible for this distribution to exist). The calculation of this type of matrix involves transforming the variables and it is during this step that a normal distribution of the variables is assumed.

  1. In response to Reviewer 1, ‘lavaan’ is not capitalized, as per the cited article.

Results

Tables

  1. In response to Reviewer 1, the contact variable and MHS data in Table 2 have been corrected and a note has been included with information about the possible value of the non-quantitative variables.

Evidence of internal structure

  1. In response to the comment by Reviewer 1: “To say Kanamori reported a general alpha for the 29 items is not indicative of unidimensionality because the general alpha could be .43, .00, .12, indicating that there may be factors that are not correlated”, we have eliminated this part of the text to ensure clarity of meaning.

  1. In response to Reviewer 1, who suggests that we report the internal consistency reliability based on the polychoric correlation matrix, we have chosen not to modify the text and to maintain Cronbach’s alpha and the categorical omega. We specifically reported the value of the scale with a poor performance for educational purposes, that is to show that when the correct matrix is used, the reliability is adequate. However, in accordance with the most up-to-date recommendations on reporting reliability coefficients, we believe that the alpha value calculated from the polychoric correlation matrix should not be systematically reported (see http://dx.doi.org/1177/0013164417727036 and http://dx.doi.org/10.6018/analesps.33.3.268401).

Evidence of validity based on the relationship with other variables

  1. In response to Reviewer 1, the contact variables were not taken into account for the multiple regression analyses as they did not surpass the significant predictive potential ( > |.3|). In this case, we are using this criterion because we are starting from an exploratory perspective. We understand that the reviewer is saying that there are findings in the literature to defend specific hypotheses. However, in the sample that we obtained, which is representative of the situation in psychology departments in Madrid, it is clear that these variables, which can be quite significant in certain contexts, are not as significant in this specific population. On the other hand, we appreciate the suggestion to report the differences in earlier analyses by contrasting the inferential hypotheses. However, more than observing the significance of the values (p<.05), we trusted the magnitude of the coefficients. Regarding the additional analyses suggested by the reviewer, we believe that they could increase the possibility of type I errors. To avoid this and, given that the differences are already modelled in the regression analysis, we have chosen to show this analysis directly.

  1. In response to the comment by Reviewer 1 regarding the multiple regression analysis: “the description of the multiple regression models is incomplete, making it very difficult to determine (i.e., were demographic variables included?) [line 432] what it means”, as noted above, the regression analysis includes all the variables that examine the values of the correlation coefficients that had an absolute value above 0.30. Therefore, these sociodemographic values were only included if they were above this threshold.

  1. In response to Reviewer 1, the sentences related to the factor structure have been clarified and the terms reviewed and modified when they were incorrect. However, we have not made any modifications related to the two factors mentioned by the reviewer, because the article does not discuss these two factors at any point. When we talk about a bifactor model, we are not referring to a 2-factor analysis, but to a factor analysis technique with a specific modelling that makes it possible to evaluate a general factor at the same time as the specific factors. See, for example: https://doi.org/10.1080/00273171.2012.715555

  1. In response to Reviewer 1, Table 4 has been corrected.

Discussion

  1. In response to Reviewer 1, the language has been reviewed and polished and the text improved to ensure clarity of meaning.

  1. In response to the comment by Reviewer 1: “The way it is worded, it sounds as if the authors fudged the data to produce psychometric reliability”, we have reworded this sentence to avoid giving this false impression. In fact, we chose to report Cronbach’s alpha and not the ordinal alpha in order to be as transparent as possible and to completely avoid giving the impression of false reliability.

  1. Regarding the comment by Reviewer 1, the discussion section has been restructured according to our response to each of the objectives listed at the end of the introduction section and in that order.

  1. In response to the comment by Reviewer 1: “Here, in the discussion the only reason apparent for rejection of the unidimensional model is the RMSEA. This might have been improved by deletion of a few items (model fitting). Then, a more parsimonous single factor scale might have been produced. Really, not much is presented as to why a 3-factor vs. 1-factor model was adopted. A comparison of chi-squares (3-factor chi subtract the 1-factor chi – was this a significant difference?)”, we appreciate this observation. However, the problem here is not that the models do not fit the data. Rather, this is a real problem of a lack of discrimination, since the questionnaire has one factor or three factors. Eliminating items because they do not reach the adequate threshold in a statistical test constitutes bad practice when questionnaires are being used. First, it limits the potential for comparison with other studies. Second, in our opinion, eliminating items to provide evidence based on the analysis of the internal structure is detrimental to other sources of validity, such as those based on the content of the item or the response process. In other words, if we eliminate items to improve the fit, we are changing the operative definition of the questionnaire and the domain that it is defining. Therefore, we believe that in this case, given that we are confirming a factor structure, we cannot use a strategy that may have been suitable in earlier phases in the development of the test.

  1. Regarding the comment made by Reviewer 1, “psychology professionals in training generally have favorable attitudes towards trans individuals” has been replaced by “psychology students …”. However, we would like to point out that in Spain, psychology undergraduates can legally work in all aspects of the psychological professions, as established by the legislation regulating the health professions, which can be consulted (in Spanish) at: Ley 44/2003, de 21 de noviembre, de ordenación de las profesiones sanitarias/Ley 5/2011, de 29 de marzo, de Economía Social. Apartado 2 de la disposición adicional sexta/Ley 33/2011, de 4 de octubre, General de Salud Pública/Real Decreto 183/2008, de 8 de febrero, por el que se determinan y clasifican las especialidades en Ciencias de la Salud y se desarrollan determinados aspectos del sistema de Formación Sanitaria Especializada).

  1. In light of the comments by Reviewer 1, some sentences that could give rise to misunderstandings have been reviewed and rewritten as recommended by the reviewer, as follows: The results related to the correlations between the variables of gender identity and sexual orientation indicated a trend, in the sense that LGB people and women had more positive attitudes towards trans people, in line with the literature on the subject [22, 54].

  1. Additionally, regarding the following comment by Reviewer 1: “Other conclusions were overreaching, as well. ‘The results related to the correlations between the variables of gender identity and sexual orientation corroborate the earlier literature, in that heterosexual individuals and men present higher levels of transphobia’ [lines 508-509]. These correlations are significant only because of the large N in this study. The correlations which the authors cite as corroborating range from r=.02 to .20 (i.e., maximum 4% of the variance based on a heavily weighted sample toward woman & heterosexual); to me, this is a contradiction of the findings in the literature”, we have decided to modify the paragraph and talk about trends, knowing that in psychology correlations > .30 are considered large according to meta-analyses (https://psycnet.apa.org/record/2014-43036-001; https://link.springer.com/article/10.1007/s12144-021-02621-7; https://journals.sagepub.com/doi/abs/10.1177/1548051815614321). Therefore, this is not a contradiction of the literature, but findings along the same line that are somewhat softer.

Conclusions

  1. In accordance with Reviewer 1, the questions related to the limitations that could result from the feminization of the sample have been expanded upon and discussed in more detail.

  1. We have also included a conclusion to characterize the analyzed sample in specific terms of transphobia.

Reviewer 2 Report

This is an excellent and fascinating paper. I do not feel qualified to judge the details of the factor analysis, but the context and background are quite strong. I have only minor criticisms:

----

Overall, the introduction is excellent, but on page 3 (101-109) it misses an important nuance in that the gender dysphoria diagnosis was argued for by many trans activists as a way to maintain a diagnosis (and access to care). The goal was to depathologize a transgender identity and instead position the discomfort/dysphoria as the thing that needed treatment.

3 Line 148 – being younger being associated with being more empathic may be a bit of a reach. I would get rid of that explanation since it could also be that being younger means greater exposure to trans individuals and thus less stigma

4 Line 199-200 “despite feeling…” does not follow from the beginning of the sentence

  1. 6 Line 271 – Were there any transgender participants or were the other 21% cisgender men?

Figure 1 is a fascinating visualization of the data.

Author Response

(The authors gave the same response as above.)

Round 2

Reviewer 1 Report

My apologies, my review is incomplete as there was not enough time to re-review the entire article.  I only was able to complete the introduction and method and I have not had an opportunity to proof-read my review.

Author Response

Dear Reviewer and Editors,

Below please find the points that the reviewer asked us to address and our response to each one. Our responses are highlighted in green, as are the changes made in the manuscript. We did not use the Microsoft Word Track Changes tool, because the new text was once again revised by a professional translator with expertise in gender and psychology.

We hope that the latest revisions are sufficiently clear and that the attached document is able to serve as the final version of the manuscript.
